# Cost-effectiveness of rapid laboratory-based mass-spectrometry diagnosis of bloodstream infection: evidence from the RAPIDO randomised controlled trial

Padraig Dixon ,[1] William Hollingworth,[2] Katie Pike,[2] Rosy Reynolds,[2] Margaret Stoddart,[3] Alasdair MacGowan[3]

¹Nuffield Department of Primary Care Health Sciences, University of Oxford, Oxford, UK
²Bristol Medical School, University of Bristol, Bristol, UK
³Southmead Hospital, Bristol, UK

**Correspondence to**
Dr Padraig Dixon;
padraig.dixon@phc.ox.ac.uk

## ABSTRACT

**Objectives and intervention** Bloodstream infection, the presence of viable micro-organisms in the blood, is a prevalent clinical event associated with substantial mortality. Patient outcomes may be improved when the causative micro-organism is identified quickly. We assessed the cost-effectiveness of rapid microbial identification by matrix-assisted laser desorption/ionisation time-of-flight (MALDI-TOF) mass spectrometry.

**Design** Economic evaluation alongside a randomised multicentre trial (RAPIDO: RAPId Diagnosis on Outcome) assessing the impact of rapid identification by MALDI-TOF spectrometry.

**Setting** Adult inpatients with bloodstream infections at seven National Health Service hospital trusts in England and Wales.

**Primary outcome** Net monetary benefit, estimated as incremental costs compared with incremental 28-day survival, of rapid identification by MALDI-TOF spectrometry compared with conventional identification.

**Methods** Patients were randomised (1:1) to receive diagnosis by conventional methods of microbial identification (conventional arm) only or by MALDI-TOF spectrometry in addition to conventional identification (RAPIDO arm).

**Results** Data from 5550 patients were included in primary analysis. Mean imputed costs in 2018/2019 prices per patient were lower by £126 in the RAPIDO arm (95% CI −£784 to £532) but the proportion of patients alive at day 28 was lower (81.4% vs 82.3%). The probability of cost-effectiveness of MALDI-TOF was <0.5 at cost-effectiveness thresholds between £20 000 and £50 000.

**Conclusions** Adjunctive MALDI-TOF diagnosis was unlikely to be cost-effective when measured as cost per death avoided at 28 days. However, the differences between arms in cost and effect were modest, associated with uncertainty and may not accurately reflect 'real-world' routine use of MALDI-TOF technology in this patient group.

**Trial registration numbers** ISRCTN97107018/UKCRN 11978.

## Strengths and limitations of this study

► We report an economic evaluation of the first randomised controlled trial of adjunctive matrix-assisted laser desorption/ionisation time-of-flight (MALDI-TOF) mass spectrometry identification of the causative micro-organism in bloodstream infection.
► Data on 5550 patients from the RAPId Diagnosis on Outcome trial were used to estimate the cost-effectiveness of MALDI-TOF in comparison to conventional microbiological methods.
► Patients were randomised on a 1:1 basis to conventional or adjunctive identification once a positive blood culture was obtained.
► We estimated the net monetary benefit, calculated as incremental costs compared with incremental 28-day survival, of rapid identification by adjunctive MALDI-TOF spectrometry compared with conventional identification.
► We calculated the cost-effectiveness of the intervention in subgroups defined by the clinical significance of the infection.

micro-organisms in the bloodstream. These infections are both prevalent and clinically significant. There are estimated to be approximately 1.2 million annual episodes of bloodstream infection in Europe, 500 000 in North America[1] and 100 000 in England and Wales.[2] Estimates of overall mortality range from 15% to 25% at 30 days postinfection to almost 50% at 12–36 months after infection.[3–6]

Rapid identification of the causative microbial pathogen may be associated with improved patient outcomes.[7 8] The RAPIDO trial assessed the impact of laboratory-based RAPId Diagnosis on Outcome of bloodstream infections in hospitalised adult patients at seven National Health Service (NHS) Hospital Trusts in England and Wales.[2] Rapid diagnosis was by matrix-assisted laser desorption/ionisation time-of-flight (MALDI-TOF) mass spectrometry applied

## BACKGROUND

Bloodstream infection refers to the presence, confirmed by laboratory testing, of viable

to machine-positive blood cultures. Here, we report the results of an economic evaluation of this trial, from an NHS perspective. Its purpose was to estimate the cost-effectiveness of using MALDI-TOF technology in addition to conventional microbiological techniques compared with conventional approaches alone.

## METHODS

Trial methods were described in MacGowan et al[2] and are summarised here.

### Design

RAPIDO was a multicentre prospective randomised (1:1) non-blinded parallel-group trial comparing two approaches to identification of the causative microorganism(s) of bloodstream infection in hospitalised adult patients at seven centres in England and Wales. The primary outcome was 28-day survival, and the two approaches were MALDI-TOF spectrometry in addition to conventional microbiological culture ('RAPIDO' arm) or conventional culture only ('conventional' arm).

### Setting and participants

Adult patients aged ≥18 years, admitted to hospital for NHS care, and with a blood sample culture-positive for bacteria or fungi were potentially eligible for inclusion, whether or not the organisms were considered clinically significant. Patients were not eligible if they were on an end-of-life pathway, had been previously randomised in the study, were prisoners or young offenders in the custody of the prison service, if the attending physician deemed them unsuitable, or if the positive blood culture entered the diagnostic pathway 'out of hours' when both MALDI-TOF and conventional identification methods were not equally available for use. Patients were randomised on a 1:1 basis to either the conventional or RAPIDO arm.

In order to initiate rapid diagnosis quickly, the study design required prompt randomisation when blood cultures flagged positive, so it was necessary to seek consent retrospectively. Research nurses approached patients for consent when they were sufficiently recovered and had capacity, in the opinion of both the clinical team and the research nurse. If a patient did not have capacity, but was thought likely to regain it, then nurses waited for capacity to return. Otherwise, a relative or close friend of the patient was approached as a consultee, if available. If patients with capacity were discharged to independent living before consent could be obtained, consent was sought by post.

### Patient and public involvement

The public and patient panel involvement group for microbiology at North Bristol NHS Trust was consulted on study design and the material given to patients.

### Intervention

Samples in both arms of the trial were tested by the conventional methods in routine use at the microbiology laboratory of the centre concerned, starting with incubation in a blood culture machine. Each laboratory's standard operating procedures defined the choice of appropriate biochemical tests and antimicrobial panels, depending on all the information about the organism available at the time.

In the RAPIDO arm, samples were first also tested by MALDI-TOF mass spectrometry, which is a relatively new technology for the identification of microbial organisms.[9] Identification may be achieved within minutes, a much shorter time than for conventional identification.[9] Microbial material was tested on Bruker Microflex MALDI-TOF mass spectrometers running Realtime Classification software V.3.1 with database V.3 (V.4 from February 2014; Bruker Daltonik GmbH, Bremen, Germany).

### Data collection and outcomes

Research nurses and laboratory staff collected data on paper data capture forms for later entry to a web-based database. Detailed data collection lasted from day 0 to day 7 after blood sampling, or discharge or death if earlier, and continued until day 28 for the key outcomes of death, discharge and *Clostridioides difficile* infection, and for laboratory data if necessary.

Key clinical data relevant to the economic evaluation included dates of admission, blood sampling (date 0), discharge and death, allowing calculation of duration of stay both before and up to 28 days after the onset of bloodstream infection. All relevant antimicrobial prescriptions were recorded from day 0 to day 7 including drug names, doses, routes and frequencies of administration, and the number of doses actually taken on each day. Ward specialty was recorded for the ward where the patient spent most of each day up to day 7.

For the economic analysis, the trial's survival outcome was expressed as the proportion of patients alive at 28 days, so that the cost-effectiveness results could be interpreted as the incremental cost per percentage change in the proportion of patients alive at 28 days.

### Measurement and valuation of resource use

The economic analysis took a health system (ie, NHS) secondary-care perspective for costs. The time horizon for the economic analysis was up to 28 days. Costs and outcomes were therefore not discounted. Costs were first calculated in 2012/2013 prices to reflect the structure of relative costs within the NHS during the first year of participant recruitment into the RAPIDO randomised controlled trial (RCT). These costs were then inflated to 2018/2019 levels. Costs from 2012/2013 to 2015/2016 were inflated using the hospital and community health services index.[10] This index was replaced in 2016 by the NHS cost inflation index which was used to inflate prices from 2015/2016 to the 2018/2019 prices. Online supplemental material contains results for the uninflated 2012/2013 price levels.

The measured components of NHS costs in each arm were: diagnostic testing (reflecting differences in

the technology randomly allocated for sample diagnosis); length of stay for initial admission and ward type (reflecting different levels of intensity of clinical support) and antimicrobials prescribed. Further details are available in online supplemental material.

### Diagnostic testing

Patients in both arms of the study had conventional diagnostic blood testing. The costs of the conventional approach to diagnosis are 'bundled' within NHS Reference Cost and tariffs categories for hospital admissions. To avoid double counting of these costs, the costs of conventional diagnosis were not separately calculated. The unit cost of identifying an organism directly from a machine-positive blood sample using MALDI-TOF was calculated using a microcosting exercise, described further in online supplemental material.

### Length of stay and ward type

Data on the specialty of the ward on which patients were located was recorded up to the seventh day after the positive blood sample was taken. Unit costs by ward type are not provided in national data sources such as NHS Reference Costs.[11] The closest analogue in NHS Reference Costs is that of 'Service Description', which groups together related procedures. The coded ward specialties were therefore matched to the closest category of 'Service Description' contained in NHS Reference Costs.

For each Service Description, we calculated a per-day cost as the average of the costs for relevant currency codes (which combine patients with similar cost implications) weighted by the frequency of codes as reported in NHS Reference Costs. We also accounted for the remuneration of hospitals according to the length of stay of patients, and for differences in elective and non-elective care. Further details are provided in online supplemental information.

We estimated ward costs between day 7 (the last day at which ward type was recorded) and day 28 (the point at which the primary outcome of the RCT was measured) by a simple extrapolation. This involved assuming that, for those patients known to survive to at least day 7, that the day 7 ward type was the ward type on which patients were located until the earliest of discharge, death or day 28. We

assess the sensitivity of the results to this assumption by comparing the primary outcome to costs at 7 days.

### Antimicrobial use

Antimicrobial drugs administered were costed to 2012/2013 prices using the British National Formulary[12] and then inflated as with other costs to 2018/2019. A per-patient, per-day antimicrobial cost was calculated from the recorded prescriptions and number of doses taken each day.

### Analytical methods

We adopted an intention-to-treat approach to analysis, in which all costs and outcomes were analysed according to the diagnostic pathway to which participants' samples were randomised, rather than the pathway actually followed. All analysis was conducted using Stata V.15.1 (StataCorp).

### Imputation of missing data

The amount of missing data was modest for patients who provided consent. Mortality data at 28 days was available for all but two patients. Information on allocation was complete. Data necessary to cost ward stays was incomplete in 4% of cases, and in 12% of cases for antimicrobials. Logistic regression analysis confirmed that for each cost category this missingness was unrelated to randomised allocation; ORs (95% CIs) for missing ward and antimicrobial data were 1.09 (0.80 to 1.49) and 0.93 (0.79 to 1.09), respectively. We used mean imputation, stratified by allocation, in order to include these data points in the complete case analysis.

Our base-case analysis used multiple imputation. Of the 5550 patients in the analysis population, 19.5% (n=1082) were eligible but unapproached survivors. Only very limited baseline information was available for these patients, in accordance with the ethical approvals received, but their mortality outcome was known. Our base-case analysis used multiple imputation to estimate the 28-day and 7-day costs of the 1082 unapproached survivors.

Multiple imputation by chained equations was implemented in Stata V.15.1 using the –ice– command.[13 14] The imputation model was stratified by trial arm and included

**Table 1** Costs in available cases

| Mean cost | Control (N=2271) | RAPIDO (N=2197) | Difference (95% CI)* | |
|---|---|---|---|---|
| Intervention cost | – | £7 | +£7 | |
| Antimicrobial cost | £272 | £292 | +£18 | (−£6 to £46) |
| 7-day ward costs | £3805 | £3757 | −£49 | (−£182 to £85) |
| Total 7-day costs | £4077 | £4055 | −£22 | (−£163 to £112) |
| 28-day ward costs | £9325 | £9282 | −£43 | (−£557 to £471) |
| Total 28-day costs | £9597 | £9580 | −£17 | (−£537 to £503) |

*CIs around mean differences calculated from unadjusted linear regression.
RAPIDO, RAPid Diagnosis on Outcome.

**Table 2** Costs following multiple imputation (based on 30 imputed datasets of N=5550)

| Mean cost | Control (N=2810) | RAPIDO (N=2740) | Difference (95% CI)* | |
|---|---|---|---|---|
| Total 7-day costs | £3667 | £3576 | −£82 | (−£321 to £157) |
| Total 28-day costs | £8253 | £8139 | −£114 | (−£773 to £545) |

*CIs around mean differences calculated from unadjusted linear regression.
RAPIDO, RAPId Diagnosis on Outcome.

all variables for which there was complete information on the 5550 analysis sample patients (centre, sex, age and consent status) and total cost at 7 and 28 days for available cases. We also included the 28-day mortality outcome. We assumed that the two patients for whom the mortality outcome was censored before day 28 had died by day 28. Predictive mean matching[13] was used to allow for non-normal distributions of the cost variables. Costs were imputed at the level of 7-day and 28-day costs, rather than for the underlying disaggregated components of these costs. The number of imputations (n=30) was selected to be larger than 100 times the proportion of missing data.[13] We reflected variation within and between the imputed datasets in the analysis using the methods described by Faria *et al*.[15]

### Cost-effectiveness analysis
Cost-effectiveness analyses were conducted using seemingly unrelated regressions, in which the outcomes of NHS costs and the proportion of patients alive at 28 days were calculated jointly. We obtained estimates of the mean difference between arms and their SEs from these regressions, which we used in calculation of the net monetary benefit (NMB) of the intervention compared with conventional identification. In the absence of a survival or mortality-specific threshold, we estimated NMB at a range of threshold values (£5000, £10 000, £20 000, £30 000 and £50 000 per death avoided at 28 days). To quantify uncertainty, we calculated CIs around point estimates of net benefit, and constructed cost-effectiveness acceptability curves.

We undertook a number of sensitivity analyses. We calculated net benefit excluding the group of eligible but unapproached survivor patients. We also calculated cost-effectiveness at 7 days (rather than 28) with and without this group of patients in order to assess whether our findings in the base-case were substantially affected by our extrapolation of ward costs beyond day 7 for participants surviving to this point.

### Subgroup analysis
We undertook one prespecified subgroup analysis in order to examine the clinical significance of the infection episode. A positive blood culture may reflect the presence of pathogenic organisms multiplying in the patient's bloodstream (clinically significant infection), or an incidental contamination of the blood sample during blood sampling or laboratory processing (not clinically significant). One may imagine relatively higher near-term costs and potentially worse survival outcomes if the infection is clinically significant. The subgroup analysis offers a test of this hypothesis.

We followed the regression-based methods for subgroup analysis set out by Willan *et al*[16] by introducing a factorial subgroup/allocation interaction into the cost and outcome equations. In addition to calculating the probability of intervention cost-effectiveness for each subgroup, we also inspected the p value associated with the interaction term in each regression for evidence of effect modification by subgroup. This analysis necessarily excluded the unapproached survivor group for whom information on clinical significance was unavailable.

## RESULTS
A total of 14 298 samples were presented for screening as first machine-positive samples from adult patients during the study period. From this total, 5670 samples were excluded as occurring out-of-hours and the remaining 8628 samples were randomised to either RAPIDO (n=4312) or conventional identification (n=4316). Excluding those who were ineligible or declined consent resulted in an analysis population of 5550 patients (2740 RAPIDO, 2810 conventional). An unexpectedly large group of patients survived to at least day 28 but were not approached for consent because they lacked capacity and no suitable consultee could be found for

**Table 3** Costs and outcome: base-case analysis with imputation (N=5550)

| | Control | RAPIDO | Difference (95% CI) | |
|---|---|---|---|---|
| Mean 28-day NHS costs | £ 8259 | £8113 | −£126 | (−£784 to £532) |
| 28-day survival* | 0.823 | 0.814 | −0.009 | (−0.029 to 0.011) |

*Survival measured as the proportion of patients alive at day 28.
NHS, National Health Service; RAPIDO, RAPId Diagnosis on Outcome.

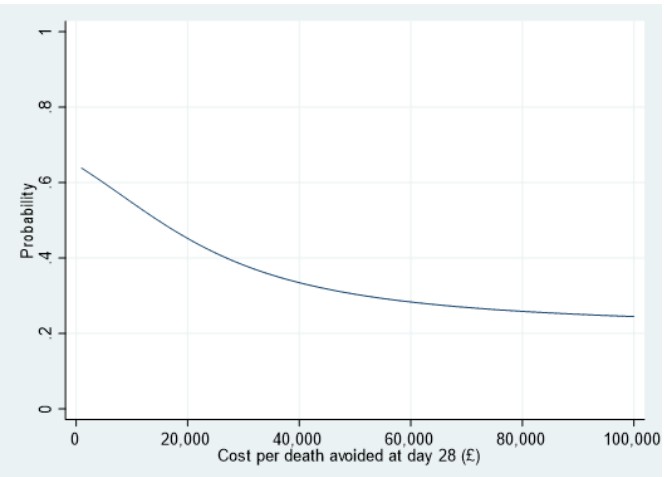

**Figure 1** Cost-effectiveness acceptability curve for base-case.

them. The analysis population of 5550 included 1082 eligible but unapproached survivors (543 RAPIDO, 539 conventional).

## Outcomes

A slightly higher proportion of patients died by 28 days in the RAPIDO group (18.5% or 508/2470) than in the conventional group (17.7% or 497/2810). The HR (calculated from Cox proportional hazards regression) was 1.05 (95% CI 0.93 to 1.19, p=0.42). Median time to discharge (up to 28 days) was 15 days in both arms (HR=0.98; 95% CI 0.90 to 1.06).

There was a limited deviation from the protocol with respect to diagnostic pathway: the correct diagnostic pathway was not followed for 6.1% of patients in the intervention arm (133/2196) and 2.1% in the control arm (48/2271).

## Costs

Costs from available cases, after mean imputation but before multiple imputation, are presented in table 1. Costs are similar between arms, with the intervention having slightly lower total costs (the sum of intervention, antimicrobial and ward costs) at 7 and 28 days.

Estimated costs following multiple imputation are presented in table 2.

**Table 4** Cost-effectiveness: base-case analysis with imputation (N=5550)

| Threshold* | Net monetary benefit (95% CI) | | Probability of cost-effectiveness |
|---|---|---|---|
| £5000 | £83 | (−£567 to £733) | 0.59 |
| £10 000 | £40 | (−£625 to £706) | 0.55 |
| £20 000 | −£45 | (−£783 to £692) | 0.45 |
| £30 000 | −£131 | (−£984 to £721) | 0.38 |
| £50 000 | −£303 | (−£1460 to £855) | 0.30 |

*Threshold value=28-day cost per death avoided at 28 days.

## Cost-effectiveness and sensitivity analysis

In the base-case imputed analysis, estimated mean costs per patient were lower in the RAPIDO arm (mean difference −£126; 95% CI −£784 to £532), and the proportion of patients alive at day 28 was also lower (81.4% vs 82.3%, see table 3). Ward costs, including the costs of conventional microbiological testing, constituted 97% of total costs in each arm. Most of the remaining 3% of total cost was attributable to antimicrobial costs. The estimated per-patient cost of diagnosis using MALDI-TOF constituted a negligible proportion of overall per-patient mean costs in the intervention arm.

The probability of the RAPIDO intervention being cost-effective declines with increasing threshold values of cost per death avoided at 28 days, as shown in figure 1, table 4.

Tables 5 and 6 report the results of the various sensitivity analyses. These analyses, expressed as NMB (with associated 95% CIs), do not differ substantially from the base-case results. Estimating costs at seven rather than 28 days did not alter the overall cost-effectiveness conclusions.

## Subgroup analysis

Tables 7 and 8 present the results of the subgroup analysis comparing clinically significant and clinically non-significant episodes of bloodstream infection. Statistical tests for interaction showed no evidence of a subgroup effect (p value for interaction in the cost seemingly unrelated regression equation=0.32, p value in the outcome seemingly unrelated regression equation=0.66), and estimates of the difference between conventional and RAPIDO diagnosis in both outcome and costs were broadly similar for the clinically significant and clinically non-significant subgroups.

## DISCUSSION

Bloodstream infections are significant, prevalent clinical events associated with substantial morbidity,[1] mortality[17] and medical cost.[18] There are an estimated 1.2 million episodes of bloodstream infection and 157 000 associated deaths per year in Europe.[1] Identification of the aetiological agent is a critical step in the treatment of bloodstream infection.

We performed a within-trial economic evaluation of MALDI-TOF diagnostic technology for the rapid identification of the causative microbial agent in hospitalised patients with bloodstream infection, excluding patients with cultures not positive for growth. The trial's primary outcome of 28-survival was consistent with no difference between conventional and adjunctive MALDI-TOF identification. The economic analysis showed that the intervention was not likely to be cost-effective, measured using incremental cost and incremental 28-day survival. The subgroup analysis suggested that there were no differences in the cost-effectiveness of MALDI-TOF when accounting for the clinical significance of the infection.

However, the differences between arms were modest and associated with considerable uncertainty. It is

**Table 5** Sensitivity analysis: costs and outcome

| | | Excluding unapproached survivors, N=4468, cost at 28 days | | Excluding unapproached survivors, N=4468, cost at 7 days | | Including unapproached survivors, N=5550, cost at 7 days | |
|---|---|---|---|---|---|---|---|
| NHS costs mean (95% CI) | Control | £9604 | (£9243 to £9967) | £4079 | (£3982 to £4117) | £3669 | (£3500 to £3836) |
| | RAPIDO | £9572 | (£9204 to £9441) | £4053 | (£3953 to £4153) | £3574 | (£3378 to £3770) |
| | Difference | –£33 | (–£549 to £484) | –£26 | (–£166 to £113) | –£95 | (–£358 to £168) |
| 28-day survival mean (95% CI) | Control | 0.78 | (0.76 to 0.80) | 0.78 | (0.76 to 0.80) | 0.82 | (0.81 to 0.84) |
| | RAPIDO | 0.77 | (0.75 to 0.79) | 0.77 | (0.75 to 0.79) | 0.81 | (0.80 to 0.83) |
| | Difference | –0.01 | (–0.04 to 0.01) | –0.01 | (–0.04 to –0.01) | –0.01 | (–0.03 to 0.01) |

NHS, National Health Service; RAPIDO, RAPId Diagnosis on Outcome.

important therefore to reflect on whether the use of this technology outside trial conditions might alter the conclusions of the within-trial evaluation. One consideration is that a higher MALDI-TOF throughput of machine-positive blood samples would reduce the overall cost per sample in the intervention arm. A value for the number of samples likely to be encountered in routine use was not included in the unit cost calculation in this study because of the exclusion criteria used in the trial: for example, it considered only samples from patients aged 18 and over. A reduction in the direct cost of MALDI-TOF would lower the intervention cost towards that of conventional diagnosis, but would not change patient outcomes.

Our economic evaluation did not calculate the cost of per-sample of conventional identification separately, since such costs are bundled into the ward stay costs and their inclusion would have amounted to double counting. By contrast, the intervention arm involved the use of MALDI-TOF in addition to conventional diagnosis, and hence the per-sample costs of MALDI-TOF are incremental to costs in the control arm. However, as MALDI-TOF has been increasingly adopted in routine practice, experience shows that it is not, in fact, used as an adjunct to conventional approaches, but largely displaces them. In addition, its widespread adoption for use with samples from much more common infections (eg, urinary tract infections) reduces its per-sample costs overall.

As a rough indication of the possible magnitudes of displacement that could be involved specifically for blood cultures, MALDI-TOF offered a usable identification of some 83% of first-bottle samples in the trial, and few of these samples would merit conventional identification in addition. Of the remaining 17% of samples, it is likely that, after further culture to produce colonial isolates, most would be successfully tested by MALDI-TOF without recourse to conventional biochemical methods.

Other considerations suggest that the incremental cost of MALDI-TOF compared with conventional identification could be smaller in 'real-world' contexts than that identified in the RAPIDO trial. Longer-term reductions in capital, labour and consumable inputs could not be measured within the period of trial-follow-up and are not reflected in the economic evaluation. For example, the MALDI-TOF process requires less physical space in the laboratory compared with conventional approaches, and a substantial long-term switch to the new technology would reduce the capital costs of microbiology laboratories, as would reductions in the cost of MALDI-TOF machines that may come from wider use and greater market competition. Changes in the workflow using MALDI-TOF reduce the time required from laboratory staff to complete an identification, meaning that results can be supplied significantly faster to clinical staff on wards (a median of 35.6 hours after taking the

**Table 6** Sensitivity analysis: cost-effectiveness

| | Excluding unapproached survivors, N=4468, cost at 28 days | | | Excluding unapproached survivors, N=4468, cost at 7 days | | | Including unapproached survivors, N=5550, cost at 7 days | | |
|---|---|---|---|---|---|---|---|---|---|
| Threshold* | NMB | (95% CI) | PCE | NMB | (95% CI) | PCE | NMB | (95% CI) | PCE |
| £5000 | –£29 | (–£533 to £475) | 0.46 | –£35 | (–£206 to £136) | 0.34 | £52 | (–£222 to £326) | 0.65 |
| £10 000 | –£91 | (–£611 to £430) | 0.37 | –£97 | (–£359 to £165) | 0.23 | £9 | (–£314 to £332) | 0.52 |
| £20 000 | –£214 | (–£841 to £414) | 0.25 | –£220 | (–£706 to £267) | 0.19 | –£77 | (–£551 to £398) | 0.38 |
| £30 000 | –£337 | (–£1135 to £461) | 0.20 | –£343 | (–£1067 to £381) | 0.18 | –£163 | (–£817 to £492) | 0.31 |
| £50 000 | –£583 | (–£1799 to –£634) | 0.17 | –£589 | (–£1796 to £618) | 0.17 | –£334 | (–£1373 to £705) | 0.26 |

*Threshold value=cost per death avoided at 28 days.
NMB, net monetary benefit (mean (95% CI)); PCE, probability of cost-effectiveness at given threshold.

**Table 7** Subgroup analysis: costs and outcome

| | | All (unapproached survivors excluded) N=4468 | | Clinically significant episodes only N=3010 (67%) | | Clinically non-significant episodes only N=1458 (33%) | |
|---|---|---|---|---|---|---|---|
| 28-day NHS costs mean (95% CI) | Control | £9604 | (£9243 to £9967) | £9456 | (£8991 to £9922) | £9451 | (£8786 to £10 116) |
| | RAPIDO | £9572 | (£9204 to £9441) | £9618 | (£9134 to £10 102) | £9047 | (£8399 to £9696) |
| | Difference | −£33 | (−£549 to £484) | £161 | (−£469 to £791) | −£404 | (−£1308 to £501) |
| 28-day survival mean (95% CI) | Control | 0.78 | (0.76 to 0.80) | 0.78 | (0.76 to 0.79) | 0.80 | (0.77 to 0.83) |
| | RAPIDO | 0.77 | (0.75 to 0.79) | 0.77 | (0.75 to 0.79) | 0.78 | (0.75 to 0.81) |
| | Difference | −0.01 | (−0.04 to 0.01) | −0.01 | (−0.04 to 0.02) | −0.02 | (−0.06 to 0.02) |

NHS, National Health Service; RAPIDO, RAPId Diagnosis on Outcome.

blood sample using MALDI-TOF compared with 54.5 hours using conventional methods in the RAPIDO trial, p<0.0001 from a Wilcoxon test given violation of the proportional hazards assumption for this outcome).

Against this, it is important to note that ward costs accounted for 97% of all secondary care costs, and the absence of a significant effect of MALDI-TOF diagnosis in reducing the length of stay and improving survival is a central conclusion of the trial—and one that merits analysis in future research (discussed below). Without evidence for improvements in these outcomes, reductions in the costs of MALDI-TOF diagnosis are plausible but may not materially alter the conclusions of our analysis.

### Strengths and limitations
#### Strengths
To our knowledge, this is the first economic evaluation conducted alongside a randomised clinical trial evaluating MALDI-TOF for rapid identification as an addition to conventional microbiological and biochemical methods in bloodstream infection. There has been much observational evidence on the effects of MALDI-TOF technology on non-health outcomes such as time to identification, but there is a lack of trial-based evidence on the costs and cost-effectiveness of using MALDI-TOF in clinical contexts.[19]

The cost-effectiveness analysis was conducted on a large dataset offering individual-level cost data on thousands of patients. The costing of the intervention itself was supported by a time and motion observational study conducted at one of the trial sites, while ward and antimicrobial data were valued using nationally representative data sources.

#### Limitations
Mortality data was censored at 28 days, and information on ward type (by far the biggest cost driver) at 7 days. We lacked the information necessary to examine longer-term costs and outcomes, although there is little reason to suspect that survival curves would diverge after 28 days to alter the primary survival outcome of the trial.

The cost analysis depended on a matching of the ward descriptions reported in the trial dataset to 'Service Descriptions' in NHS Reference Costs. This matching

process was performed 'blind' to allocation, but the analysis of 28-day costs required an extrapolation from 7 days to the earliest of death, discharge or the 28-day end-point. Unobserved changes in ward type after day 7 could change the estimated costs used in the base-case, although no substantial difference was observed when comparing 28-day and 7-day analyses.

We did not have access to primary care records so our analysis was limited to a secondary care NHS perspective; that is, hospital resource use. In practice, because of the magnitude of per-day ward costs, it is unlikely that accounting for other health system costs would have a material impact on our findings, given that the mortality outcome favoured conventional diagnosis.

Patients who were unable to consent for themselves and for whom no consultee was available comprised 19.5% (n=1082) of the 5550 patients in the analysis population, and no data beyond the mortality outcome, allocation, age and sex was available for them. However, apart from this group of patients, and the absence of ward-specific data between days 8 and 28, the amount of missing data in key cost drivers was limited.

The multiple imputation model estimated that these 1082 patients had lower mean costs than the other 4468 participants. Is this plausible? On the one hand, costs should be expected to diverge between the two groups given that the 1082 patients who did not provide consent are all known to have survived for at least 28 days. Thus, the lower costs generated by the imputation model may reflect a population less acutely ill than the other participants. On the other hand, prolonged survival without discharge would give rise to higher costs than early death during the trial period.

These considerations complicate assessments of direction of the biases in the available case data. However, the conclusions that emerge from the multiple imputation results, the available case results, and the various sensitivity analyses are similar in identifying considerable uncertainty around the cost-effectiveness of MALDI-TOF in this patient group.

Finally, the economic evaluation was limited to identifying the cost-effectiveness of the intervention and did not identify the mechanisms that gave rise to the survival

**Table 8** Subgroup analysis: cost-effectiveness at 28 days

| Threshold value* | All (unapproached survivors excluded) N=4468 | | | Clinically significant episodes only N=3010 (67%) | | | Clinically non-significant episodes only N=1458 (33%) | | |
|---|---|---|---|---|---|---|---|---|---|
| | NMB | (95% CI) | PCE | NMB | (95% CI) | PCE | NMB | (95% CI) | PCE |
| £5000 | −£29 | (−£533 to £475) | 0.46 | −£206 | (−£853 to £441) | 0.27 | £301 | (−£629 to £1231) | 0.74 |
| £10 000 | −£91 | (−£611 to £430) | 0.37 | −£250 | (−£947 to £446) | 0.24 | £198 | (−£803 to £1199) | 0.65 |
| £20 000 | −£214 | (−£841 to £414) | 0.25 | −£339 | (−£1206 to £527) | 0.22 | −£9 | (−£1254 to £1237) | 0.49 |
| £30 000 | −£337 | (−£1135 to £461) | 0.20 | −£428 | (−£1521 to £664) | 0.22 | −£215 | (−£1785 to £1355) | 0.39 |
| £50 000 | −£583 | (−£1799 to −£634) | 0.17 | −£606 | (−£2222 to £1010) | 0.23 | −£627 | (−£2949 to £1694) | 0.30 |

*Threshold value=cost per death avoided at 28 days.
NMB, net monetary benefit (mean (95% CI)); PCE, probability of cost-effectiveness at given threshold.

outcomes in the trial. Evidence from the analysis of secondary clinical outcomes in the RAPIDO trial indicates that time to provision of microbiological identification to the ward was significantly shorter in the RAPIDO arm and there was weak evidence of longer time to initiation of appropriate antimicrobial therapy in the RAPIDO arm (median 24.0 vs 13.0 hours, p=0.056 based on a Cox proportional hazards model). However, there was no significant difference between arms in other secondary outcomes: time to providing Gram stain and antimicrobial susceptibility results to the ward; time to resolution of fever (up to 7 days) or discharge (up to 28 days); *C. difficile* incidence (to 28 days); in-hospital antimicrobial consumption (to 7 days) or the proportion of patients remaining on broad-spectrum therapy at 7 days.[2]

### Future research
Future research could examine the mechanisms by which mortality outcomes may differ between MALDI-TOF and conventional diagnosis. Analysis of length of stay and survival in observational study designs in US contexts by Huang *et al*[20] and Perez *et al*[21] found beneficial effects of MALDI-TOF when used in conjunction with antimicrobial stewardship programmes, hinting at organisational changes that may be needed to exploit the faster identification offered by MALDI-TOF.[22] However, this is context-specific, as in other health systems, such as the NHS in which the present trial was conducted, bacteraemia consultation teams are routine and involved in care at an early stage.

### CONCLUSION
Overall, the evidence from the RAPIDO trial suggests that the use of MALDI-TOF as an adjunct to conventional microbial identification is unlikely to offer value when its incremental costs are compared with 28-day incremental survival. It is plausible that the costs of MALDI-TOF in 'real-world' routine use may well be lower than those measured during the RAPIDO trial, and savings can also be expected as it would displace much conventional testing.

**Acknowledgements** We are very grateful to all patients, healthcare professionals and NHS staff who contributed time and effort to make the RAPIDO trial possible. We are grateful to administrative staff at trial sites for support with participant recruitment, data entry and trial administration. The RAPIDO trial was designed and delivered in collaboration with Clinical Trials and Evaluation Unit Bristol (CTEU), a UKCRC Registered Clinical Trials Unit in receipt of National Institute for Health Research CTU support funding. RR acknowledges support from the NIHR Health Protection Research Unit in Behavioural Science and Evaluation at University of Bristol.

**Contributors** PD and WH: conducted the economic analysis. PD: wrote the first draft of this manuscript. PD, WH, KP, RR, MS, AM: reviewed, commented and edited the manuscript. AM: chief investigator for the RAPIDO trial.

**Funding** This report summarises independent research funded by the National Institute for Health Research (NIHR) under its Programme Grant for Applied Research (PGfAR RP-PG-0707-10043).

**Competing interests** None declared.

**Patient consent for publication** Not required.

**Ethics approval** The RAPIDO trial was approved by the NRES Committee South West - Frenchay on 20 March 2012, reference 12/SW/003.

**Provenance and peer review** Not commissioned; externally peer reviewed.

**Data availability statement** Data are available upon reasonable request. Permission to share individual-level data was not obtained from trial participants. Requests for summary level data should be directed to AM

**ORCID iD**
Padraig Dixon http://orcid.org/0000-0001-5285-409X

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
