## [Reviewer comments · BMJ Open]

ARTICLE DETAILS

TITLE (PROVISIONAL)	Cost-effectiveness of rapid laboratory-based mass-spectrometry diagnosis of bloodstream infection: Evidence from the RAPIDO randomized controlled trial
AUTHORS	Dixon, Pdraig; Hollingworth, William; Pike, Katie; Reynolds, Rosy; Stoddart, Margaret; MacGowan, Alasdair

VERSION 1 – REVIEW

REVIEWER	Mustaffa, Nazri Universiti Sains Malaysia - Kampus Kesihatan
REVIEW RETURNED	30-Sep-2020

GENERAL COMMENTS	The manuscript is well-written, with the results discussed appropriately. Strengths and limitations of the study are also recognised as well as commented on. However, I am unfamiliar with many of the statistical methods used; these may need further review
---

REVIEWER	Tanabe, Katsuyuki Okayama University Graduate School of Medicine Dentistry and Pharmaceutical Sciences
REVIEW RETURNED	13-Dec-2020

GENERAL COMMENTS	In this manuscript, authors demonstrated that matrix-assisted laser desorption/ionization time-of-flight (MALDI-TOF) mass spectrometry-mased microbial identification was unlikely to be cost-effective to avoid short-term death compared with conventional bacterial culture in a randomized controlled trial including 5,500 patients with bloodstream infection. The subject of study is interesting, and data analysis appears to be appropriate. However, there are some concerns in this study. The reviewer's comments are described as follows. Major points: 1. A critical disadvantage of microbial identification by MALDI-TOF mass spectrometry is that it cannot accompany antibiotics susceptibility tests. On the other hand, an advantage of MALDI-TOF mass spectrometry is that it may identify fastidious and slow growing bacteria that cannot be identified by conventional bacterial culture. However, patients without negative culture test were excluded from this study according to the study design. Therefore, these advantage and disadvantage of MALDI-TOF mass spectrometry were not reflected in the cost-effectiveness in this study.2. Since there are no data regarding the patient characteristics and the severity of infectious diseases at the time of randomization in both conventional group and MALDI-TOF group, it is difficult to estimate the effects of MALDI-TOF mass spectrometry on the prognosis and medical costs. Taken together with the above
--

	comment, whether MALDI-TOF mass spectrometry was more cost-effective compared with conventional culture test could not be concluded based on the results. Minor points:  1. Authors should describe the name of the ethical committee and the approval number assigned by the committee. 2. Authors should describe the equipment of MALDI-TOF mass spectrometry in detail. In particular, whether they used MALDI Biotyper or VITEK MS, or both, in this study should be disclosed. 3. Authors should describe the statistical methods to determine the significant differences between two groups and calculate the p values in the Method section.
--	--

REVIEWER	Badrick, Tony Royal College of Pathologists of Australasia, QAP
REVIEW RETURNED	23-Apr-2021

GENERAL COMMENTS	This is a well designed, implemented and analysed trial of the cost-effectiveness of using MALDI-TOF identification of bloodstream infectious agents. The paper is well written and the discussion scholarly.  1. Page 10, line 27. Expand on the statement 'Cost-effectiveness parameters were calculated parametrically from the output of seemingly unrelated regressions.' 2. Page 11, line 53. Is there a reason why there was the reported difference in whether or not the diagnostic pathway was followed? Was this because it was not normal practice? 3. Supplementary Data: Line 25 a missing Table reference and in Line 29 there is an error.
--

REVIEWER	Mao, Wenhui Duke Global Health Institute
REVIEW RETURNED	02-May-2021

GENERAL COMMENTS	The methods were stated clearly and analysis were properly performed.
---

REVIEWER	Bogavac-Stanojević, Nataša University of Belgrade
REVIEW RETURNED	26-Jul-2021

GENERAL COMMENTS	The Manuscript entitled "Cost-effectiveness of rapid laboratory-based mass-spectrometry diagnosis of bloodstream infection: Evidence from the RAPIDO randomized controlled trial" contains specific issues that limit comprehensive understanding of the obtained results. For this reason, the current recommendation to the Editorial board would be to invite Authors to make revision prior to considering the Manuscript for publication. Comments and suggestions are the following: Despite some explanations given in the supplementary documents and the limitation of the study, it is not clear why the researchers opted for two scenarios presented in the Manuscript. Namely, in the discussion section, the authors stated that the Maldi TOF method is used instead of the conventional method in routine practice and not as an additional method to the conventional one. "However, as MALDI-TOF has been increasingly adopted in routine practice, experience shows that it is not, in fact, used as an adjunct to conventional approaches, but largely displaces them." They noted that the Maldi TOF method is fast, so it is introduced into
---

	regular practice. Nevertheless, the researchers chose another scenario that is not characteristic of routine practice. Both strategies are based on all patient testing by the conventional method, and one arm is followed by Maldy TOF testing. It is better to examine cost – effectiveness of diagnostic procedures accepted in practice, even on a smaller number of patients, than unaccepted diagnostic procedures. The study design presented in the Manuscript could not test the basic assumption that the turnaround time (TAT) for the Maldy TOF method is better than the conventional method. In such cases, the therapy should be more successful, which is the fundamental goal of any diagnostic procedure. Also, from this study's design, it is unclear what was added value of conventional testing followed by the Maldy TOF method compared to conventional testing alone. Namely, it is not indicated whether these patients received more effective therapy due to additional MALDI testing. The authors tried to explain this in the study's limitations. The authors stated, "Finally, the economic evaluation was limited to identifying the cost-effectiveness of the intervention and did not identify the mechanisms that gave rise to the survival outcomes in the trial." However, without this analysis, the presented results are inadequate because there is no evidence that survival and calculated costs depend on the diagnostic procedure. Introduction section The authors are recommended to indicate why the new combination of methods is better than the previous one? Why is the therapy's effectiveness expected to be better due to the diagnostic procedure and why the costs arising from the specific diagnostic procedure are expected to be different? In the discussion section, the authors should consider whether the hypothesis is correct. If there is no difference in effectiveness between strategies, why did the authors not analyse data by the cost minimization method? It would be helpful for readers to present patient selection graphically and explain the difference in outcomes in more detail if they were observed per the protocol and with the intention to treat analysis. The tables present the results of a few sensitivity analyzes that were not mentioned in the discussion section. What is the purpose of such analyzes and what can be concluded from them? Of particular interest is the subgroup analysis, which presents the NMB value results for persons with clinically significant and nonsignificant positive results. Again, no comments related to subgroup analyses were obtained.
--	---

VERSION 1 – AUTHOR RESPONSE

Reviewer: 1

Dr. Nazri Mustaffa, Universiti Sains Malaysia - Kampus Kesihatan

Comments to the Author:

The manuscript is well-written, with the results discussed appropriately. Strengths and limitations of the study are also recognised as well as commented on. However, I am unfamiliar with many of the statistical methods used; these may need further review

RESPONSE: None required.

Reviewer: 2

Dr. Katsuyuki Tanabe, Okayama University Graduate School of Medicine Dentistry and
Pharmaceutical Sciences

Comments to the Author:

In this manuscript, authors demonstrated that matrix-assisted laser desorption/ionization time-of-flight (MALDI-TOF) mass spectrometry-based microbial identification was unlikely to be cost-effective to avoid short-term death compared with conventional bacterial culture in a randomized controlled trial including 5,500 patients with bloodstream infection. The subject of study is interesting, and data analysis appears to be appropriate. However, there are some concerns in this study. The reviewer's comments are described as follows.

Major points:

COMMENT 1. A critical disadvantage of microbial identification by MALDI-TOF mass spectrometry is that it cannot accompany antibiotics susceptibility tests. On the other hand, an advantage of MALDI-TOF mass spectrometry is that it may identify fastidious and slow growing bacteria that cannot be identified by conventional bacterial culture. However, patients without negative culture test were excluded from this study according to the study design. Therefore, these advantage and disadvantage of MALDI-TOF mass spectrometry were not reflected in the cost-effectiveness in this study.

RESPONSE: Note that it is patients with cultures not positive for growth that were excluded post-randomization, not "patients without negative culture test".

Our results should therefore be considered as conditional on the exclusion of patients with cultures not positive for growth. We have made this clear in the Discussion section of the manuscript as follows.

"We performed a within-trial economic evaluation of MALDI-TOF diagnostic technology for the rapid identification of the causative microbial agent in hospitalised patients with bloodstream infection, excluding patients with cultures not positive for growth."

COMMENT: 2. Since there are no data regarding the patient characteristics and the severity of infectious diseases at the time of randomization in both conventional group and MALDI-TOF group, it is difficult to estimate the effects of MALDI-TOF mass spectrometry on the prognosis and medical costs. Taken together with the above comment, whether MALDI-TOF mass spectrometry was more cost-effective compared with conventional culture test could not be concluded based on the results.

RESPONSE: We disagree. In expectation, across all randomizations, we may expect covariates to be balanced between the arms of a trial. This process permits unbiased estimates of the relative effect of conventional+MALDI-TOF diagnosis compared to conventional diagnosis alone in relation to all parameters required to calculate the cost-effectiveness of MALDI-TOF diagnosis.

Minor points:

COMMENT 1. Authors should describe the name of the ethical committee and the approval number assigned by the committee.

RESPONSE: This information was provided in the original submission as follows

"Ethics: The RAPIDO trial was approved by the NRES Committee South West - Frenchay on 20 March 2012, reference 12/SW/003."

COMMENT: 2. Authors should describe the equipment of MALDI-TOF mass spectrometry in detail. In particular, whether they used MALDI Biotyper or VITEK MS, or both, in this study should be disclosed.

RESPONSE: We have now added this information as follows:

“Microbial material was tested on Bruker Microflex MALDI-TOF mass spectrometers running Realtime Classification software version 3.1 with database version 3 (version 4 from February 2014; Bruker Daltonik GmbH, Bremen, Germany).”

COMMENT 3. Authors should describe the statistical methods to determine the significant differences between two groups and calculate the p values in the Method section.

RESPONSE: We did not characterise any differences in the main analysis on a definition of “significant” differences between arms for our main analysis. Instead, we used the net benefit approach to estimate the probability that the intervention was likely to be cost-effective. We made two uses of p-values. One was in relation to the interaction terms in the subgroup analysis and in a comparison of the time to supply identification information to ward staff. We have added text to the Methods section as follows:

“In addition to calculating the probability of intervention cost-effectiveness for each subgroup, we also inspected the p-value associated with the interaction term in each regression for evidence of effect modification by subgroup.”

The second use was in relation to time-to-event outcomes mentioned in the discussion. Time to initiation of appropriate antimicrobial therapy was based on a Cox proportional hazards model. Time to provision of microbiological identify was identified using a Wilcoxon test given violations of the proportional hazards assumption. These models were stratified by research centre.

We have added text to the Discussion on these points as follows.

“Changes in workflow using MALDI-TOF reduce the time required from laboratory staff to complete an identification, meaning that results can be supplied significantly faster to clinical staff on wards (a median of 35.6 hours after taking the blood sample using MALDI-TOF compared to 54.5 hours using conventional methods in the RAPIDO trial, $p < 0.0001$ from a Wilcoxon test given violation of the proportional hazards assumption for this outcome).”

“Evidence from the analysis of secondary clinical outcomes in the RAPIDO trial indicates that time to provision of microbiological identification to the ward was significantly shorter in the RAPIDO arm and there was weak evidence of longer time to initiation of appropriate antimicrobial therapy in the RAPIDO arm (median 24.0 versus 13.0 hours, $p = 0.056$ based on a Cox proportional hazards model).”

Reviewer: 3

Dr. Tony Badrick, Royal College of Pathologists of Australasia

Comments to the Author:

COMMENT This is a well designed, implemented and analysed trial of the cost-effectiveness of using MALDI-TOF identification of bloodstream infectious agents. The paper is well written and the discussion scholarly.

1. Page 10, line 27. Expand on the statement 'Cost-effectiveness parameters were calculated parametrically from the output of seemingly unrelated regressions.'

RESPONSE: We have modified this text as follows;

“Cost-effectiveness analyses were conducted using seemingly unrelated regressions, in which the outcomes of NHS costs and the proportion of patients alive at 28 days were calculated jointly. We obtained estimates of the mean difference between arms and their standard errors from these

regressions, which we used in calculation of the net monetary benefit of the intervention compared to conventional identification.”

COMMENT: 2. Page 11, line 53. Is there a reason why there was the reported difference in whether or not the diagnostic pathway was followed? Was this because it was not normal practice?

RESPONSE: The reasons for this are not clear. A protocol deviation was coded if the correct diagnostic path was not followed for at least one sample bottle. It is possible that this occurred slightly more often in the intervention arm because of the additional steps required compared to conventional practice, but we do not have data to test this hypothesis.

COMMENT: 3. Supplementary Data: Line 25 a missing Table reference and in Line 29 there is an error.

RESPONSE: We have corrected these errors as follows.

Revised Line 25: “Descriptions and currency codes in NHS Reference Costs as shown in Table A1.”

Revised Line 29: “An example calculation is shown in Table A2”

Reviewer: 4

Dr. Wenhui Mao, Duke Global Health Institute

Comments to the Author:

The methods were stated clearly and analysis were properly performed.

RESPONSE: None required.

Reviewer: 5

Dr. Nataša Bogavac-Stanojević, University of Belgrade

Comments to the Author:

COMMENT The Manuscript entitled “Cost-effectiveness of rapid laboratory-based mass-spectrometry diagnosis of bloodstream infection: Evidence from the RAPIDO randomized controlled trial” contains specific issues that limit comprehensive understanding of the obtained results. For this reason, the current recommendation to the Editorial board would be to invite Authors to make revision prior to considering the Manuscript for publication. Comments and suggestions are the following:

Despite some explanations given in the supplementary documents and the limitation of the study, it is not clear why the researchers opted for two scenarios presented in the Manuscript. Namely, in the discussion section, the authors stated that the Maldi TOF method is used instead of the conventional method in routine practice and not as an additional method to the conventional one. “However, as MALDI-TOF has been increasingly adopted in routine practice, experience shows that it is not, in fact, used as an adjunct to conventional approaches, but largely displaces them.”

They noted that the Maldi TOF method is fast, so it is introduced into regular practice. Nevertheless, the researchers chose another scenario that is not characteristic of routine practice. Both strategies are based on all patient testing by the conventional method, and one arm is followed by Maldi TOF testing. It is better to examine cost – effectiveness of diagnostic procedures accepted in practice, even on a smaller number of patients, than unaccepted diagnostic procedures.

RESPONSE: The RAPDIO randomized controlled trial compared the impact of different means of identifying the causative organism in bloodstream infection. The trial was motivated by the observation that increasing the speed of diagnosis for these infections may improve patient outcomes.

Improvements in outcomes using more rapid diagnosis have also been identified, in for example, the earlier use of appropriate chemotherapy. MALDI-TOF using direct extraction from blood cultures offers an opportunity to improve the time to identification, and therefore may have improved outcomes. Note also that it MALDI-TOF was begun alongside or even ahead of the very first steps of conventional testing (Gram stain and microscopy) in the intervention arm. It was not the case that MALDI-TOF “followed” conventional testing.

An RCT is the most appropriate study design to assess how MALDI-TOF performs in this respect compared to conventional identification. Current practice is best informed by robust RCTs such as RAPIDO, and it is in the nature of RCTs that some novel intervention is compared to current practice. That some uses of MALDI-TOF have entered in routine practice does not abolish the argument for stringent tests of its effectiveness; if anything, it reinforces the need to assess whether current or emerging practice will support the interests of patients at an acceptable cost.

COMMENT: The study design presented in the Manuscript could not test the basic assumption that the turnaround time (TAT) for the Maldy TOF method is better than the conventional method. In such cases, the therapy should be more successful, which is the fundamental goal of any diagnostic procedure. Also, from this study's design, it is unclear what was added value of conventional testing followed by the Maldy TOF method compared to conventional testing alone. Namely, it is not indicated whether these patients received more effective therapy due to additional MALDI testing. The authors tried to explain this in the study's limitations. The authors stated, "Finally, the economic evaluation was limited to identifying the cost-effectiveness of the intervention and did not identify the mechanisms that gave rise to the survival outcomes in the trial." However, without this analysis, the presented results are inadequate because there is no evidence that survival and calculated costs depend on the diagnostic procedure.

RESPONSE: For the avoidance of doubt, the principal hypothesis tested in the RAPIDO RCT was on its use as an adjunct to conventional identification in relation to 28-day survival. Analysis of outcomes under random allocation to either of the two trial arms (conventional only, or conventional plus MALDI-TOF identification) permits a robust and unbiased test of whether survival, cost, and cost-effectiveness differed according to diagnostic procedure in this patient group. The main trial paper is cited as MacGowan et al in our submission and provides detail about study design, processes and outcomes.

COMMENT Introduction section

The authors are recommended to indicate why the new combination of methods is better than the previous one?

RESPONSE: See responses to previous comments – we cite references (7 and 8) in the Introduction that indicated why more rapid identification of the causative agent may support better outcomes. We tested this hypothesis in the RCT.

COMMENT Why is the therapy's effectiveness expected to be better due to the diagnostic procedure and why the costs arising from the specific diagnostic procedure are expected to be different? In the discussion section, the authors should consider whether the hypothesis is correct. If there is no difference in effectiveness between strategies, why did the authors not analyse data by the cost minimization method?

RESPONSE: See previous responses – the RCT is a direct test of whether the new diagnostic procedure might be more effective than conventional methods.

We have amended the Discussion to re-emphasize the null findings of the RCT as follows “The trial’s primary outcome of 28-day survival was consistent with no difference between conventional and adjunctive MALDI-TOF identification. “

As explained in Dakin and Wordsworth(Health Economics, 2013 Jan;22(1):22-34. doi: 10.1002/hec.1812), cost-minimization is only appropriate in a very narrow set of circumstances, since it will bias estimates of uncertainty, which will cause over- or under-estimates of the probability that treatment under investigation is cost-effective. None of the very few circumstances in which cost-minimization is likely to be appropriate applied in the context of our study.

COMMENT It would be helpful for readers to present patient selection graphically and explain the difference in outcomes in more detail if they were observed per the protocol and with the intention to treat analysis.

RESPONSE: We have now included a copy of the trial’s CONSORT diagram in supplementary material. This is reproduced here for reference.

Notes to figure: Flow of patients. a Ineligible samples include 111 rapid diagnosis and 125 conventional samples that were randomized in error, and 872 rapid diagnosis and 828 conventional that were randomized correctly but met postrandomization exclusion criteria. b Unapproached survivors are eligible patients who could not be approached for consent, usually because of lack of capacity and inability to identify a consultee. They are included in mortality analysis (only) as 28-day survivors.

We did not plan or undertake per-protocol analysis. Instead, we analyzed the data on an ITT basis.

COMMENT The tables present the results of a few sensitivity analyzes that were not mentioned in the discussion section. What is the purpose of such analyzes and what can be concluded from them? Of particular interest is the subgroup analysis, which presents the NMB value results for persons with clinically significant and nonsignificant positive results. Again, no comments related to subgroup analyses were obtained.

RESPONSE: We have altered the Methods section to describe our approach to subgroup analysis, and their relevance to the wider analysis.

Our pre-specified subgroup analysis assessed whether there may have been differences in cost-effectiveness by the clinical significance of the identified organism. This accounted for any differences in unapproached survivors. Patients were randomised in a 1:1 ratio once a positive blood culture was identified, but prior to assessment of eligibility or seeking consent. This aspect of the study design was necessary given that many of the individuals randomized into the trial were severely ill.

Unapproached survivors are patients, eligible for the trial, who could not be approached for consent because they lacked capacity or because a consultee could not be identified.

One can imagine that rapid identification of the causative organism may lead to greater costs if further investigations and procedures are necessary for a clinically significant organism compared to an organism that is not clinically significant. Moreover, one may imagine a greater relative impact of mortality in identifying a significant organism more rapidly than with conventional identification.

However, neither of these effects appears to have an impact on this trial. This finding is notable since it indicates that the cost-effectiveness of MALDI_TOF as a diagnostic intervention does not depend on the clinical significance of the identified organism.

We have now added text to the Methods as follows: “One may imagine relatively higher near-term costs and potentially worse survival outcomes if the infection is clinically significant. The subgroup analysis offers a test of this hypothesis.”

And to the Discussion as follows “The subgroup analysis suggested that there were no differences in the cost-effectiveness of MALDI-TOF when accounting for the clinical significance of the infection. “

VERSION 2 – REVIEW

REVIEWER	Bogavac-Stanojević, Nataša University of Belgrade
REVIEW RETURNED	23-Sep-2021
GENERAL COMMENTS	I am glad to note that most of the comments submitted by the reviewers have been mastered and that authors have made efforts to correct them. The current form of the article is clearer, the messages are explicit and their importance is emphasized.